# Choroidal Hyperreflective Nodules Detected by Infrared Reflectance Images Are a Diagnostic Criterion for Neurofibromatosis Type 1 Patients Excluding Those with High Myopia

**DOI:** 10.3390/diagnostics13071348

**Published:** 2023-04-04

**Authors:** Marta Orejudo de Rivas, Javier Mateo Gabás, Miguel Ángel Torralba Cabeza, Olivia Esteban Floría, Raquel Herrero Latorre, Eva Núñez Moscarda, Julia Aramburu Clavería, Guillermo Pérez Rivasés, Javier Ascaso Puyuelo

**Affiliations:** 1Department of Ophthalmology, Lozano Blesa University Clinic Hospital, 50009 Zaragoza, Spain; 2Aragon Health Research Institute (IIS Aragon), 50009 Zaragoza, Spain; 3Department of Internal Medicine, Lozano Blesa University Hospital, 50009 Zaragoza, Spain; 4School of Medicine, Department of Surgery, University of Zaragoza, 50009 Zaragoza, Spain

**Keywords:** neurofibromatosis type 1, choroidal nodules, Lisch nodules

## Abstract

Neurofibromatosis type 1 (NF1) is one of the central nervous system’s most common autosomal dominant conditions. The diagnosis is based on the clinical diagnostic criteria and/or a molecularly confirmed mutation in the NF1 gene. This study investigated the possibility of substantiating choroidal nodules as a diagnostic criterion for the disease, including patients affected with and without high myopia. A cross-sectional study was carried out in 60 eyes of 30 adult patients diagnosed with NF1. A total of 30 healthy individuals of equivalent age and sex served as control. The Spectralis HRA+OCT MultiColor (Heidelberg Engineering GmbH, Heidelberg, Germany) evaluated the presence of choroidal abnormalities with near-infrared reflectance imaging. Secondly, the presence of iridian Lisch nodules was evaluated by slit lamp examination. Near-infrared reflectance imaging showed the presence of choroidal hyperreflective nodules in 83% of the patients diagnosed with NF1, while these choroidal abnormalities were not observed in any control subject. The patients diagnosed with NF1 associated with high myopia were the only ones who did not present the characteristic choroidal disorders. Therefore, when excluding patients diagnosed with high myopia, choroidal nodules were more frequent than Lisch nodules in a statistically significant proportion. Hyperreflective nodules detected by near-infrared reflectance imaging are as regular as Lisch nodules or even significantly more frequent when excluding high myope patients. Our observation of the mutual exclusion of choroidal hyperreflective nodules and high myopia in the NF1 patients seems a novel and interesting remark.

## 1. Introduction

Neurofibromatosis type 1 is an autosomal dominant disorder involving aberrant proliferation of multiple tissues of neural crest origin. It is principally associated with cutaneous, neurologic, and orthopedic manifestations. Most epidemiological studies have reported a prevalence between 1/3000 and 1/6000. It is a very heterogeneous disorder, with multisystem involvement in most cases, affecting individuals variably. NF1 reduces life expectancy due to an increased risk of malignancy [1,2].

The diagnosis of NF1 is based on the clinical diagnostic criteria established by the National Institutes of Health (NIH) and/or a molecularly confirmed mutation in the NF1 gene (complete sequencing of the coding region of the most relevant genes for clinical practice in solid adult tumors was tested when patients presented with an unusual phenotype or an incomplete clinical picture).

The NIH clinical diagnostic criteria for NF1 [3,4] have high specificity and sensitivity for most patients. Ophthalmological manifestations of NF1 that are included in the established diagnostic criteria are Lisch nodules, optic glioma, and a distinctive osseous lesion (sphenoid wing dysplasia) [5]. Legius et al. [6] revised the diagnostic criteria for NF1 in 2021, incorporating significant developments and creating new criteria for Legius syndrome (LGSS) to differentiate the two conditions. They proposed an initiative to update the diagnostic criteria for NF1, including choroidal abnormalities as an ophthalmologic criterion because of its high specificity, sensitivity, and ability to differentiate from LGSS.

Few reports have shown the presence of choroidal hyperreflective nodules [1,2,3,5,6,7,8,9,10,11,12,13,14,15,16,17,18,19,20,21,22,23]. The reason is that they are undetectable with conventional ophthalmoscopic fundus examination [5]. The retinal pigment epithelium (RPE) contains melanin that blocks the passage of visible light to the choroid. Infrared light penetrates the RPE better than visible light [7] and allows observing choroidal abnormalities.

In a sample of NF1 patients, the study aimed to investigate whether the presence of choroidal nodules detected by NIR (near-infrared reflectance) is a valuable criterion for diagnosis in patients with and without high myopia.

## 2. Materials and Methods

All patients signed the informed consent. The protocol of this study was approved by the Research Ethics Committee of the Autonomous Community of Aragón (Spain).

Between September 2021 and May 2022, 60 eyes of 30 white Caucasian adult patients (eighteen female, twelve male; mean age 57 years) diagnosed with NF1 using the National Institutes of Health (NIH) criteria were studied. So far, the diagnosis relies on clinical diagnosis, and genetic testing is optional when the diagnosis has already been established [24]. We also examined 60 eyes of 30 healthy control subjects. Each subject underwent a general ophthalmologic examination, including an accurate slit lamp examination searching for Lisch nodules. Patients with NF1 were referred from the department of Internal Medicine (where only patients over 18 are checked) to the department of Ophthalmology of the Lozano Blesa University Hospital in Zaragoza (Spain) as a part of a screening program to detect ocular pathology in NF1 patients. Patients under eighteen years old were excluded from the study since it has been found that choroidal abnormalities tend to increase with patient age [8], and it was intended to find a homogeneous sample, as well as those with the presence of media opacities and retinal pathology.

Near-infrared reflectance at 815 nm and red-free (RF) 488 nm images of the posterior pole and mid-periphery of the retina were taken of each patient and monitored with a confocal scanning laser ophthalmoscope (Spectralis HRA+OCT, Heidelberg Engineering, Heidelberg, Germany) (Figure 1) [9]. The combined spectral domain optical coherence tomography (SD-OCT) system allows for simultaneous recording with precise alignment of OCT and topographic images (NIR, RF). Using this technology, which is non-invasive, fast, and abundantly used in routine exams, a good correlation of the topographic image with the morphological changes in the retina and choroid of the macular area was obtained [9]. No dilating eye drops were used before the examination.

Red-free images were analyzed in a masked way by the same investigator, who searched for choroidal hyperreflective nodules that were counted, objectifying more than 10 in most patients diagnosed with NF1, and any other unusual findings. A different independent investigator repeated the examination to obtain another analysis for an agreement evaluation.

A similar procedure occurred with slit lamp examination where the principal investigator noted the presence or absence of iridian Lisch nodules (Figure 2) and took anterior segment photographs with the Topcon slit lamp imaging system (SL-D701, Topcon Healthcare). They were also counted, objectifying an average of 11 nodules in the patients diagnosed with NF1. Another independent investigator reviewed the images and repeated the analysis to obtain data for the consensus of the final results.

Data regarding refractive errors, which could affect the retinal anatomy, and consequently the detection of bright, patchy nodules by NIR also was noted. High myopia was defined as the spherical equivalent refraction of −5.0 D or more.

## 3. Results

In the interest of verifying the homogeneity of the two experimental groups (including 60 eyes of 30 Caucasian adult patients in each group) in terms of certain sociodemographic variables, the age and sex of the patients were compared. It was observed that there were no significant differences between both groups.

The frequency of choroidal hyperreflective nodules, detected by NIR, and Lisch nodules, detected by slit lamp examination, were compared between patients and control subjects by the Chi-square test. It was observed that, in the group diagnosed with NF1, 80% showed iridian Lisch nodules, while in the control group, there were no cases (Table 1).

The numbers in Table 1 and Table 2 represent the number of patients, considering that Lisch and choroidal nodules were always bilateral.

In the analysis of the comparison in the diagnosis of choroidal patches between the two experimental groups employing the Chi-square test, it was observed that 83.3% of patients diagnosed with NF1 showed the alteration, whereas, in the control group, none of them did so (Table 2).

When comparing whether there are significant differences in the prevalence of choroidal patches and Lisch nodules among patients with NF1, considering those with high myopia, it is observed that there are no significant differences between the prevalence (Table 3).

When analyzing the sample of patients with NF1, patients with high myopia were not considered (five patients), it was observed that 84% of patients showed Lisch nodules compared to 100% of patients that presented choroidal nodules. The differences between prevalence were statistically significant, confirming the higher prevalence of choroidal patches in this new group of patients close to emmetropia (Table 4).

## 4. Discussion

The prevalence of choroidal hyper-reflective nodules may be higher than believed because asymptomatic patients without eye discomfort or blurred vision tend not to undergo ocular examinations. Moreover, patients with good vision that perform a thorough eye exam usually are subjected to an eye fundus (that does not reveal the existence of this particular finding) and are not tested with either NIR imaging or OCT.

Previous research shows that choroidal nodules in the setting of NF1 can be seen in up to 82% of NF1 patients [10]. Moramarco et al. [11] reported a prevalence of 97% of choroidal abnormalities in patients with NF1. This study reports that 83.3% of patients diagnosed with NF1 present choroidal hyperreflective nodules. The study’s limitations include the small number of patients, and the OCT with NIR, although present in many ophthalmology departments, might be absent. What is new in our study and what we want to emphasize are that our research also studies the prevalence of choroidal nodules in patients diagnosed with NF1, excluding patients with high myopia, resulting in 100% prevalence. This is a new finding that has not been previously described before in the literature. We speculate that the choroid of highly myopic patients is significantly thinner, and choroidal vascularization is considerably altered in these patients; This characteristic may have to do with the lower presence of choroidal nodules [12,13]. Considering that most of the population does not suffer from high myopia, we believe that more studies should be carried out, including this new exclusion criterion, since it has been shown to significantly increase the prevalence of choroidal nodules in patients with NF1.

The bright lesions on NIR imaging correspond to the hyperflow areas of deep choroids on OCT angiography (OCTA), indicating a rich vascular supply of these nodules. When studying the choroid in these nodules, an alteration of the morphology and thickness has been reported. The mean choroidal thickness is reduced with generalized thinning of the neuroepithelium, retinal pigmentary epithelium, and outer nuclear layer [14].

Considering the literature, the presence of choroidal nodules in NF1 does not produce a visual loss or other ophthalmological symptomatology, so it continues to be a diagnostic challenge to detect them.

Although choroidal abnormalities have already been previously included in the revised diagnostic criteria as described by Legius et al., only a consensus recommendation has been achieved; we believe that this study can provide more support to the literature to consolidate choroidal nodules as a new diagnostic criterion and can also offer a new exclusion criterion to support the evidence further already raised.

## 5. Conclusions

NIR OCT represents a non-invasive, easy-to-perform reproducible exam to detect choroidal nodules in NF1 patients. Although more studies are needed to support the evidence, the current research suggests that the frequency of appearance of these choroidal changes leads to the possibility of adding choroidal hyperreflective nodules as an additional diagnostic criterion for NF1 but only in patients without high myopia.

## Figures and Tables

**Figure 1 diagnostics-13-01348-f001:**
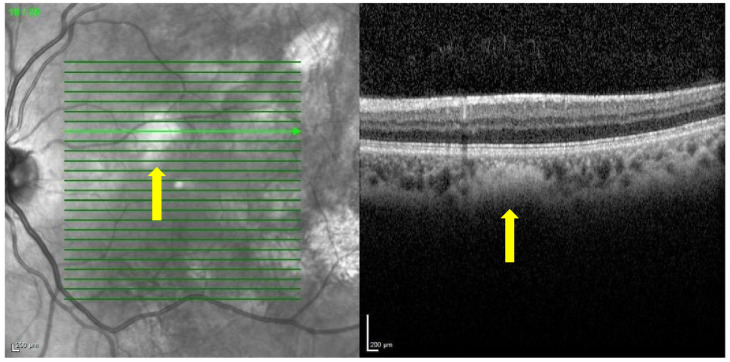
Choroidal nodules detected by Spectralis HRA+OCT. The image on the left shows the posterior pole of the retina in the form of a cube, visualized with near-infrared reflectance. The green lines are translated into horizontal slices. The section containing a choroidal nodule (yellow arrow) is selected, and the nodule segmented by OCT is shown in the image on the right.

**Figure 2 diagnostics-13-01348-f002:**
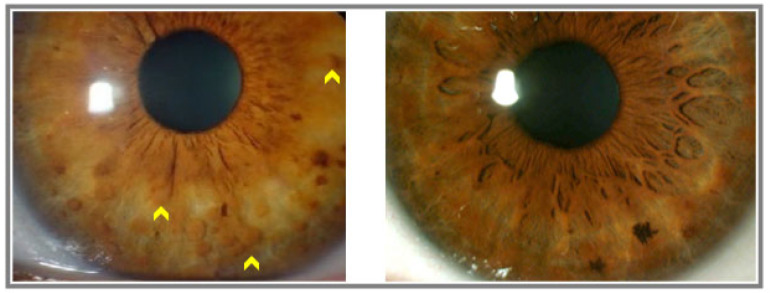
Iridian Lisch nodules detected with slit lamp examination on the left image (yellow arrow heads). Normal iris evaluated with slit lamp examination on the right image.

**Table 1 diagnostics-13-01348-t001:** Percentage of Lisch nodules according to experimental groups.

	Experimental Group		
Lisch Nodules	Control	Case	*p* ^1^
No	30 (100.0%)	6 (20%)	<0.001
Yes	0 (0.0%)	24 (80.0%)	
Absolute frecuency ^1^ Chi-square			

**Table 2 diagnostics-13-01348-t002:** Percentage of Choroidal nodules according to experimental groups.

	Experimental Group		
Choroidal Nodules	Control	Case	*p* ^1^
No	30 (100.0%)	5 (16.7%)	<0.001
Yes	0 (0.0%)	25 (83.3%)	
Absolute frecuency ^1^ Chi-square			

**Table 3 diagnostics-13-01348-t003:** Comparison of the proportion of Lisch nodules and choroidal nodules in patients with NF1.

Variable	Freq ^§^ (%)	*p* ^2^
Choroidal nodules	25 (83.3%)	0.3705
Lisch nodules	24 (80.0%)	
^2^ Z test ^‡^		

^§^ Freq: frequency. ^‡^ Z test: Zulliger test.

**Table 4 diagnostics-13-01348-t004:** Comparison of the proportion of Lisch nodules and choroidal nodules in patients with NF1 without high myopia.

Variable	Freq ^§^ (%)	*p* ^2^
Choroidal nodules	25 (100.0%)	0.0145
Lisch nodules	21 (84.0%)	
^2^ Z test ^‡^		

^§^ Freq: frequency. ^‡^ Z test: Zulliger test.

## Data Availability

The data presented in this study are available on request from the Department of Internal Medicine, Lozano Blesa University Hospital.

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
