# Peer review of "Choroidal Hyperreflective Nodules Detected by Infrared Reflectance Images Are a Diagnostic Criterion for Neurofibromatosis Type 1 Patients Excluding Those with High Myopia"

_diagnostics, 2023, doi:10.3390/diagnostics13071348_

Round 1
Reviewer 1 Report
In their study entitled “Choroidal hyperreflective nodules detected by infrared reflectance images as a potential diagnostic criterion for neurofibromatosis-1”, de Rivas and co-authors evaluated the presence of choroidal abnormalities in 60 eyes of 30 patients diagnosed with NF1 and compared it to the group of 30 healthy individuals of equivalent age and sex who served as controls. The purpose of their study was “to investigate the possibility of including choroidal nodules as a potentially new diagnostic criterion for the disease.” Their investigation “showed the presence of choroidal hyperreflective nodules in 83% of the patients diagnosed with NF1, while these choroidal abnormalities were not observed in any control subject.” Interestingly, “the patients diagnosed with Neurofibromatosis type 1 associated with high myopia were the only ones who did not present the characteristic choroidal disorders.” The authors concluded that “although more studies are needed to support the evidence, the current study suggests that choroidal hyperreflective nodules could be considered as a new diagnostic criterion for NF1.”
MAJOR PROBLEMS
1. Choroidal abnormalities are already included in the revised diagnostic NF1 criteria as described by Legius et al. in “Revised diagnostic criteria for neurofibromatosis type 1 and Legius syndrome: an international consensus recommendation” (Genetics in Medicine 2021, vol. 23, pp.1506-1513). The authors site this paper several times in their manuscript. It is unclear what is the rational for their study. However, the authors’ observation of the mutual exclusion of choroidal hyperreflective nodules and high myopia in the NF1 patients seems to be a novel and interesting observation. The authors should consider restructuring their manuscript to emphasize this finding.
2. Reported clinical data for the patients involved in the study are minimal, and the molecular testing data are absent. The manuscript could be of the higher value and of increased interest to the readers if the authors included individual demographic and clinical data (including approx. counts of the Lisch nodules and choroidal abnormalities) as well as the results of the NF1 molecular testing.
3. Both figures in the manuscript should be more informative. For instance, there are two panels in Figure 1, but there is no appropriate libeling or description of the panels in the figure’s legend. What do the horisontal lines and the arrow in the left panel designate? What the reader should perceive from the image shown in the right panel? A few most typical Lisch nodules shown in Figure 2 should be identified with arrowheads, and an image of a normal eye should be supplied for comparison.
4. What do the numbers shown in Tables 2 and 3 mean? Do they represent the number of patients or the number of eyes? It is unclear what was the number of controls. 30 or 31? The numbers in the text and the tables do not match. Were the Lisch and choroidal nodules bilateral in every patient?
5. The reason for excluding NF1 patients younger than 18 y.o. from the study should be discussed and justified.
MINOR PROBLEMS
1. In the title, the term “neurofibromatosis-1” should be replaced with “neurofibromatosis type 1”.
2. Information shown in Table 1 is obsolete and should be replaced with the revisited NF1 diagnostic criteria. The table could be placed in the supplementary materials.
3. NF1 affects multiple organs and tissues, not just the central nervous system.
4. There’s a mistake in Materials and Methods section, line 67: “Between September 2022 and May 2023… “
5. Section “6. Patents” (line 161) has no information.
Author Response
- Response to the comments of Reviewer 1:
First of all, thank you for your kind words. We will try to answer below your comments:
Major problems
- Choroidal abnormalities are already included in the revised diagnostic NF1 criteria described by Legius et al. in “Revised diagnostic criteria for neurofibromatosis type 1 and Legius syndrome: an international consensus recommendation” (Genetics in Medicine 2021, vol. 23, pp.1506-1513). The authors cite this paper several times in their manuscript. It is unclear what is the rationale for their study. However, the author’s observation of the mutual exclusion of choroidal hyperreflective nodules and high myopia in NF1 patients seems novel and interesting. The authors should consider restructuring their manuscript to emphasize this finding.
The manuscript was restructured, and the new finding was emphasized.
- Reported clinical data for the patients involved in the study are minimal, and the molecular testing data are absent. The manuscript could be of higher value and increased interest to the readers if the authors included individual demographic and clinical data (including approx. counts of the Lisch nodules and choroidal abnormalities) and the results of the NF1 molecular testing.
Clinical data of the patients was extended; molecular testing included complete sequencing of the coding region of the 56 most relevant genes for clinical practice in solid adult tumors (line 42). A few examples of genetic findings were: mutation c.4409_4410, p.Ala1470Gly, pathogenic variant c.484C>T (p.Gln162*), c.3427C>T (p.His1143Tyr), c.3916C>T (p.Arg1306), c.7389del (p.Tyr2464Ile fs*4) in the NF1 gene.
- Both figures in the manuscript should be more informative. For instance, there are two panels in Figure 1, but there is no appropriate labeling or description of the panels in the figure’s legend. What do the horizontal lines and the arrow in the left panel designate? What should the reader perceive from the image shown in the proper forum? A few most typical Lisch nodules shown in Figure 2 should be identified with arrowheads, and a picture of a routine eye should be supplied for comparison.
Figures were modified (lines 205, 222).
- What do the numbers are shown in Tables 2 and 3 mean? Do they represent the number of patients or the number of eyes? It is unclear what was the number of controls. 30 or 31? The numbers in the text and the tables do not match. Were the Lisch and choroidal nodules bilateral in every patient?
Numbers shown in tables 1 and 2 were explained (line 242). Table numbers were checked (line 241).
- The reason for excluding NF1 patients younger than 18 y.o. from the study should be discussed and justified.
The reason for excluding NF1 patients younger than 18 y.o. from the study was discussed and justified (line 128, line 131).
Minor problems
- In the title, “neurofibromatosis-1” should be replaced with “neurofibromatosis type 1”.
The title was modified.
- The information in Table 1 is obsolete and should be replaced with the revisited NF1 diagnostic criteria. The table could be placed in the supplementary materials.
Table 1 was deleted.
- NF1 affects multiple organs and tissues, not just the central nervous system.
Information concerning NF1 was extended (line 35).
- There’s a mistake in the Materials and Methods section, line 67: “Between September 2022 and May 2023…. “
The mistake was corrected (line 121).
- Section “ Patents”(line 161) has no information.
The section titled was corrected (line 317).

Reviewer 2 Report
1.Did you rule out a cytomegalovirus eye infection from the medical diagnosis?
2.From Conclusions please delete:
"together with Legius et al [6]. work"
3.Please write what is new in your study and does not require confirmation with other subsequent studies.
Author Response
- Response to the comments of Reviewer 2:
We appreciate your comments about our study. We will try to answer below your comments:
- Did you rule out a cytomegalovirus eye infection from the medical diagnosis?
We did not perform a cytomegalovirus test since the study was based on a series of patients already diagnosed with Neurofibromatosis type 1. Neither was clinical suspicion of CMV infection in any patient.
- From Conclusions, please delete:
"together with Legius et al. [6]. work"
"together with Legius et al. [6]. Work" was deleted from the conclusions.
- Please write what is new in your study and does not require confirmation with other subsequent studies.
Although choroidal abnormalities have already been previously included in the revised diagnostic criteria as described by Legius et al., only a consensus recommendation has been achieved; we believe that this study can provide more support to the literature to consolidate choroidal nodules as a new diagnostic criterion, and also offers a new exclusion criterion to support the evidence further already raised.

Reviewer 3 Report
This is an interesting but already published and well-proved idea. I did not see anything new this study will add to the current understanding.
At lease, the authors should try to expand their discussion session to provide their opinions on the similarity or difference in their study and other papers already being published. Overall, the paper cannot be accepted in this current version. Other comments:
1. The results should start with a summary of how many eyes and patients included.
2. Please speculate: why the choroidal hyperreflective nodules were not detected in high myopia? What did this indicate?
Author Response
- Response to the comments of Reviewer 3:
First of all, thank you for your kind words. We will try to answer below your comments:
- The results should start with a summary of how many eyes and patients are included.
You can find the summary of how many eyes and patients were included in lines 228-229.
- Why were the choroidal hyperreflective nodules not detected in high myopia? What did this indicate?
In response to the second demand of Reviewer 3, we speculate that the choroid of highly myopic patients is significantly thinner than that of those who are not. This characteristic may have to do with the lower presence of choroidal nodules (line 291).

Reviewer 4 Report
Thank you for the opportunity to review this interesting article.
I recommend its publication after introducing the following remarks:
1. The Authors wrote at the end of Introduction section in the purpose of the study that choroidal nodules can be detected by NIR. What is NIR? The abbreviation was not used previously, please explain what does it mean.
2. Please check Table 2 and 3. The number of controls is 31 not 30…
3. What are the limitations of the study? Please write a separate paragraph at the end of discussion section.
4. The discussion should be broaden, I am aware that there are not many publications about this issue, however is should be expanded.
Author Response
- Response to the comments of Reviewer 4:
We appreciate your comments about our study. We will try to answer below your comments:
- The Authors wrote at the end of the Introduction section the purpose of the study that NIR can detect choroidal nodules. What is NIR? The abbreviation was not used previously; please explain what it means.
The definition of NIR is now explained in line 116.
- Please check Tables 2 and 3. The number of controls is 31, not 30…
Table numbers were checked (line 241).
- What are the limitations of the study? Please write a separate paragraph at the end of the discussion section.
In response to the third demand, the study's limitations include the small number of patients and the absence of OCT with NIR in many ophthalmology departments (line 286).
- The discussion should be broadened; I know there are not many publications about this issue but is should be expanded.
The discussion was broadened.

Round 2
Reviewer 1 Report
Can be accepted in present form.
Reviewer 2 Report
NICE WORK
Reviewer 3 Report
The authors have addressed my questions.